# LEARNING A MAX-MARGIN CLASSIFIER FOR CROSS-DOMAIN SENTIMENT ANALYSIS

## ABSTRACT

Sentiment analysis is a costly yet necessary task for enterprises to study the opinions of their customers to improve their products and services and to determine optimal marketing strategies. Due to the existence of a wide range of domains across different products and services, cross-domain sentiment analysis methods have received significant attention in recent years. These methods mitigate the domain gap between different applications by training cross-domain generalizable classifiers which help to relax the need for individual data annotation for each domain. Most existing methods focus on learning domain-agnostic representations that are invariant with respect to both the source and the target domains. As a result, a classifier that is trained using annotated data in a source domain, would generalize well in a related target domain. In this work, we introduce a new domain adaptation method which induces large margins between different classes in an embedding space based on the notion of prototypical distribution. This embedding space is trained to be domain-agnostic by matching the data distributions across the domains. Large margins in the source domain help to reduce the effect of "domain shift" on the performance of a trained classifier in the target domain. Theoretical and empirical analysis are provided to demonstrate that the method is effective.

## 1 INTRODUCTION

The main goal in sentiment classification is to predict the polarity of users automatically after collecting their feedback, e.g., Amazon customer reviews. Popularity of online shopping and reviews, fueled further by the recent pandemic, provides a valuable resource for businesses to study the behavior and preferences of consumers and to align their products and services with the market demand. A major challenge for automatic sentiment analysis is that polarity is expressed using completely dissimilar terms and phrases in different domains. For example, while terms such as "fascinating" and "boring" are used to describe books, terms such as "tasty" and "stale" are used to describe food products. As a result of this discrepancy, a model that is trained for a particular domain may not generalize well in other different domains, referred as the problem of "domain gap" (Wei et al., 2018). Since generating annotated training data for all domains is expensive and time-consuming, cross-domain sentiment analysis has gained significant attention recently (Saito et al., 2018; Li et al., 2017; Peng et al., 2018; He et al., 2018; Li et al., 2018; Barnes et al., 2018; Sarma et al., 2019; Li et al., 2019; Guo et al., 2020; Xi et al., 2020; Dai et al., 2020; Lin et al., 2020). The goal in cross-domain sentiment classification is to relax the need for data annotation via transferring knowledge from another domain with annotated data to domains with unannotated data.

The above problem has been studied more broadly in the "domain adaptation" literature. A common approach for domain adaptation is to map data points from two domains into a shared embedding space to align the data distributions (Redko & Sebban, 2017). Since the embedding space would become domain-agnostic, i.e., a classifier that is trained using the source domain annotated data, will generalize in the target domain. In the sentiment analysis problem, this means that polarity of natural language can be expressed independent of the domain in the embedding space. We can model this embedding space as the output of a shared deep encoder which is trained to align the distributions of both domains at its output. This training procedure have been implemented using both adversarial learning (Pei et al., 2018; Long et al., 2018; Li et al., 2019; Dai et al., 2020), which aligns distributions indirectly, or by loss functions that are designed to directly align the two distributions (Peng et al., 2018; Barnes et al., 2018; Kang et al., 2019; Guo et al., 2020; Xi et al., 2020; Lin et al., 2020).

**Contributions:** our main contribution is to develop a new cross-domain sentiment analysis algorithm for model adaptation by introducing large margins between classes in the source domain. Our idea is based on learning a prototypical distribution for the source domain in a cross-domain embedding space which is trained to be domain-agnostic. We model this distribution as a Gaussian mixture modal (GMM). We estimate the parameters of the prototypical distribution using a subset of source samples for which the classifier is confident about its predictions. As a result, larger margins between classes are introduced in the prototypical distribution which help reducing domain gap. We then use this prototypical distribution to align the source and the target distributions via minimizing the Sliced Wasserstein Distance (SWD) (Lee et al., 2019). We draw confident random samples from this distribution and enforce the distribution of the target in the embedding matches this prototypical distribution in addition to the source distribution. We provide a theoretical proof to demonstrate that our method minimizes an upperbound for the target domain expected error. Experimental results demonstrate that our algorithm outperforms state-of-the-art sentiment analysis algorithms.

## 2 RELATED WORK

While domain adaptation methods for visual domains usually use generative adversarial networks (GANs) (Goodfellow et al., 2014) and align distributions indirectly, the dominant approach for cross-domain sentiment analysis is to design appropriate loss functions that directly impose domain alignment. The main reason is that natural language is expressed in terms of discrete values such as words, phrases, and sentences. Since this domain is not continuous, even if we convert natural language into real-valued vectors, it is not differentiable. Hence, adversarial learning procedure cannot be easily implemented for pure natural language processing (NLP) applications. Several alignment loss functions have been designed for cross-domain sentiment analysis. A group of methods are based on aligning the lower-order distributional moments, e.g., means and covariances, across the two domains, in an embedding space (Wu & Huang, 2016; Peng et al., 2018; Sarma et al., 2019; Guo et al., 2020). An improvement over these methods is to use probability distribution metrics to consider the encoded information in higher order statistics (Shen et al., 2018). Damodaran et al. (Bhushan Damodaran et al., 2018) demonstrated that using Wasserstein distance (WD) for domain alignment boosts the performance significantly in visual domain applications (Long et al., 2015; Sun & Saenko, 2016). In the current work, we rely on the sliced Wasserstein distance (SWD) for aligning distribution. SWD has been used for domain adaptation in visual domains (Lee et al., 2019).

The major reason for performance degradation of a source-trained model in a target domain stems from "domain shift", i.e., the boundaries between the classes change in the embedding space even for related domains which in turn increases possibility of misclassification. It has been argued that if a max-margin classifier is trained in the source domain, it can generalize better than many methods that try to align distributions without further model adaptation (Tommasi & Caputo, 2013). Inspired by the notion of "class prototypes", our method is based on both aligning distributions in the embedding space and also inducing larger margins between classes using the notion of "prototypical distributions". Recently, cross-domain alignment of the class prototypes has been used for domain adaptation (Pan et al., 2019; Chen et al., 2019). The idea is that when a deep network classifier is trained in a domain with annotated data, data points of classes form separable clusters in an embedding space, modeled via network responses in hidden layers. A class prototype is defined as the mean of each class-specific data cluster in the embedding space. Domain adaptation then can be addressed by aligning the prototypes across the two domains as a surrogate for distributional alignment. Following the above, our work is based on using the prototypical distribution, rather simply the prototypes, to induce maximum margin between the class-specific clusters after an initial training phase in the source domain. Since the prototypical distribution is a multimodal distribution, we can estimate it using a Gaussian mixture model (GMM). We estimate the GMM using the source sample for which the classifier is confident and use random samples with high-confident labels to induce larger margins between classes, compared to using the original source domain data.

## 3 CROSS-DOMAIN SENTIMENT ANALYSIS

Consider two sentiment analysis problems in a source domain $\mathcal{S}$ with an annotated dataset $D_{\mathcal{S}} = (\boldsymbol{X}_{\mathcal{S}}, \boldsymbol{Y}_{\mathcal{S}})$, where $\boldsymbol{X}_{\mathcal{S}} = [\boldsymbol{x}_1^s, \ldots, \boldsymbol{x}_N^s] \in \mathcal{X} \subset \mathbb{R}^{d \times N}$ and $\boldsymbol{Y}_{\mathcal{S}} = [\boldsymbol{y}_1^s, \ldots, \boldsymbol{y}_N^s] \in \mathcal{Y} \subset \mathbb{R}^{k \times N}$ and a target domain $\mathcal{T}$ with an unannotated dataset $D_{\mathcal{T}} = (\boldsymbol{X}_{\mathcal{S}})$, where $\boldsymbol{X}_{\mathcal{T}} = [\boldsymbol{x}_1^t, \ldots, \boldsymbol{x}_N^t] \in \mathcal{X} \subset$

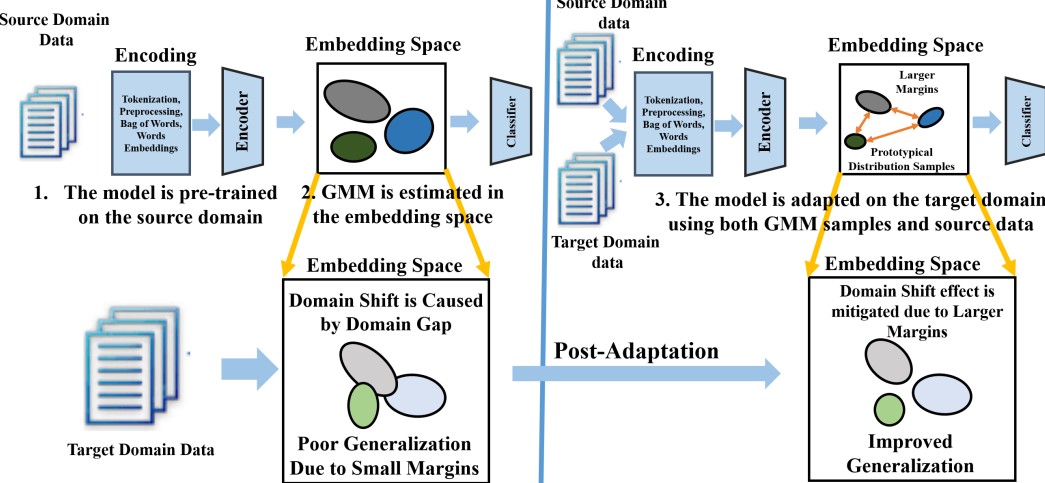

Figure 1: Architecture of the proposed cross-domain sentiment analysis framework. Left: separable clusters are formed in the embedding space after initial supervised model training in the source domain and then the prototypical distribution is estimated as a GMM. Right: random samples from the GMM with high-confident labels are used to generate a pseudo-dataset. The pseudo-dataset induces larger margins between the classes to mitigate the effect of domain shift in the target domain.

$\mathbb{R}^{d \times M}$. The real-valued feature vectors $\boldsymbol{X}_{\mathcal{S}}$ and $\boldsymbol{X}_{\mathcal{T}}$ are obtained after pre-processing the input text data using common NLP methods, e.g., bag of words or word2vec. We consider that both domains share the same type of sentiments and hence the one-hot labels $\boldsymbol{y}_i^s$ encode $k$ sentiment types, e.g., negative or positive in binary sentiment analysis. Additionally, we assume that the source and the target feature data points are drawn independent and identically distributed from the domain-specific distributions $\boldsymbol{x}_i^s \sim p_S(\boldsymbol{x})$ and $\boldsymbol{x}_i^t \sim p_T(\boldsymbol{x})$, respectively, where $p_T(\boldsymbol{x}) \neq p_S(\boldsymbol{x})$.

Given a family of parametric functions $f_\theta : \mathbb{R}^d \to \mathcal{Y}$, e.g., deep neural networks with learnable parameters $\theta$, and considering an ideal labeling function $f(\cdot)$, e.g., $\forall (\boldsymbol{x}, \boldsymbol{y}) : \boldsymbol{y} = f(\boldsymbol{x})$, the goal is to search for the optimal predictor model $f_{\theta^*}(\cdot)$ in this family for the target domain. This model should have minimal expected error for sentiment analysis, i.e., $\theta^* = \arg\min_\theta\{e_\theta\} = \arg\min_\theta\{\mathbb{E}_{\boldsymbol{x}^t \sim p_T(\boldsymbol{x})}(\mathcal{L}(f(\boldsymbol{x}^t), f_\theta(\boldsymbol{x}^t)))\}$, where $\mathcal{L}(\cdot)$ is a proper loss function and $\mathbb{E}(\cdot)$ denotes the expectation operator. Since the target domain data is unlabeled, the naive approach is to estimate the optimal model using the standard empirical risk minimization (ERM) in the source domain:

$$\hat{\theta} = \arg\min_\theta\{\hat{e}_\theta(\boldsymbol{X}_{\mathcal{S}}, \boldsymbol{Y}_{\mathcal{S}}, \mathcal{L})\} = \arg\min_\theta\{\frac{1}{N}\sum_i \mathcal{L}(f_\theta(\boldsymbol{x}_i^s), \boldsymbol{y}_i^s)\} \tag{1}$$

Given a large enough labeled datasets in the source domain, ERM model generalizes well in the source domain. The source-trained model may also perform much better than chance in the target domain, given cross-domain. However, its performance will degrade in the target domain compared to its performance in the source domain due to existing distributional discrepancy between the two domains, since $p_S \neq p_T$. Our goal is to benefit from the encoded information in the unlabeled target domain data and adapt the source-trained classifier $f_{\hat{\theta}}$ to generalize in the target domain. We use the common approach of reducing the domain gap by mapping data into a shared embedding space.

We consider that the predictor model $f_\theta(\cdot)$ can be decomposed into a deep encoder $\phi_{\boldsymbol{v}}(\cdot) : \mathcal{X} \to \mathcal{Z} \subset \mathbb{R}^p$ and a classifier $h_{\boldsymbol{w}}(\cdot) : \mathcal{Z} \to \mathcal{Y}$ such that $f_\theta = h_{\boldsymbol{w}} \circ \phi_{\boldsymbol{v}}$, where $\theta = (\boldsymbol{w}, \boldsymbol{v})$. $\mathcal{Z}$ is an embedding space which is modeled by the encoder responses at its output. We assume that the classes have become separable for the source domain in this space after an initial training phase (see Figure 1, left). If we can adapt the source-trained encoder network such that the two domains share similar distributions $\mathcal{Z}$, i.e., $\phi(p_S)(\cdot) \approx \phi(p_T)(\cdot)$, the embedding space would become domain-agnostic. As a result, the source-trained classifier network will generalize with similar performance in the target domain. A number of prior cross-domain sentiment analysis algorithms use this strategy, select a proper probability distribution metric to compute the distance between the distributions $\phi(p_S(\boldsymbol{x}^s))$

and $\phi(p_\mathcal{T}(\boldsymbol{x}^t))$, and then train the encoder network to align the domains via minimizing this distance:

$$\hat{\boldsymbol{v}}, \hat{\boldsymbol{w}} = \arg \min_{\boldsymbol{v},\boldsymbol{w}} \frac{1}{N} \sum_{i=1}^{N} \mathcal{L}\big(h_{\boldsymbol{w}}(\phi_{\boldsymbol{v}}(\boldsymbol{x}_i^s)), \boldsymbol{y}_i^s\big) + \lambda D\big(\phi_{\boldsymbol{v}}(p_\mathcal{S}(\boldsymbol{X}_\mathcal{T})), \phi_{\boldsymbol{v}}(p_\mathcal{T}(\boldsymbol{X}_\mathcal{T}))\big), \qquad (2)$$

where $D(\cdot, \cdot)$ denotes a probability metric to measure the domain discrepancy and $\lambda$ is a trade-off parameter between the source ERM and the domain alignment term. In this work, we use SWD (Lee et al., 2019) to compute $D(\cdot, \cdot)$ in equation 2. Using SWD has three advantages. First, SWD can be computed efficiently compared to WD based on a closed form solution of WD distance in 2D. Second, SWD can be computed using the empirical samples that are drawn from the two distributions. Finally, SWD possesses a non-vanishing gradient even when the support of the two distributions do not overlap (Bonnotte, 2013; Lee et al., 2019). Hence SWD is suitable for deep learning problems which are normally solved using first-order gradient-based optimization techniques, e.g., Adam.

While methods based on variations of equation 2 are effective to reduce the domain gap to some extent, our goal is to improve upon the baseline obtained by equation 2 by introducing a loss term that increases the margin between classes in the target domain (check the embedding space in Figure 1, right). By doing so, our goal is to mitigate the negative effect of domain shift in the target domain.

## 4    LEARNING MAX-MARGIN CLASSIFIERS

---

**Algorithm 1** SAUM² $(\lambda, \tau)$

---
1: **Initial Training**:
2:   **Input:** source dataset $\mathcal{D}_\mathcal{S} = (\boldsymbol{X}_\mathcal{S}, \boldsymbol{Y}_\mathcal{S})$,
3:     **Training on the Source Domain:**
4:     $\hat{\theta}_0 = (\hat{\boldsymbol{w}}_0, \hat{\boldsymbol{v}}_0)$
5:       $= \arg\min_\theta \sum_i \mathcal{L}(f_\theta(\boldsymbol{x}_i^s), \boldsymbol{y}_i^s)$
6:   **Prototypical Distribution Estimation:**
7:     Use equation 4 and estimate $\alpha_j, \boldsymbol{\mu}_j, \Sigma_j$
8: **Model Adaptation**:
9:   **Input:** target dataset $\mathcal{D}_\mathcal{T} = \boldsymbol{X}_\mathcal{T}$
10:   **Pseudo-Dataset Generation:**
11:     $\hat{\mathcal{D}}_\mathcal{P} = (\mathbf{Z}_\mathcal{P}, \mathbf{Y}_\mathcal{P}) =$
12:     $([\boldsymbol{z}_1^p, \ldots, \boldsymbol{z}_N^p], [\boldsymbol{y}_1^p, \ldots, \boldsymbol{y}_N^p])$, where:
13:         $\boldsymbol{z}_i^p \sim \hat{p}_J(\boldsymbol{z}), 1 \le i \le N_p$
14:         $\boldsymbol{y}_i^p = \arg\max_j\{h_{\hat{\boldsymbol{w}}_0}(\boldsymbol{z}_i^p)\}$,
15:         $\max\{h_{\hat{\boldsymbol{w}}_0}(\boldsymbol{z}_i^p)\} > \tau$
16: **for** $itr = 1, \ldots, ITR$ **do**
17:     draw data batches from $\mathcal{D}_\mathcal{S}, \mathcal{D}_\mathcal{T}$, and $\mathcal{D}_\mathcal{P}$
18:     Update the model by solving equation 6
19: **end for**

---

Our idea for increasing margins between the classes is based on using an intermediate prototypical distribution in the embedding space. We demonstrate that this distribution can be used to induce larger margins between the classes. To this end, we consider that the classifier subnetwork consists of a softmax layer. This means that the classifier should become a maximum *a posteriori* (MAP) estimator after training to be able to assign a membership probability to a given input feature vector. Under this formulation, the model will generalize in the source domain if after supervised training of the model using the source data, the input distribution is transformed into a multi-modal distribution $p_J(\cdot) = \phi_{\boldsymbol{v}}(p_\mathcal{S})(\cdot)$ with $k$ modes in the embedding space (see Figure 1, left). Each mode of this distribution corresponds to one type of sentiments. The mean for each of these modes corresponds to the notion of "class prototype" in the prior works (Pan et al., 2019; Chen et al., 2019). Following a similar terminology, we refer to this distribution as the prototypical distribution. The geometric distance between the modes of prototypical distribution corresponds to the margins between classes. If we test the source-trained model in the target domain, the boundaries between class modes will change due to the existence of "domain shift", i.e., $\phi_{\boldsymbol{v}}(p_\mathcal{T})(\cdot) \ne \phi_{\boldsymbol{v}}(p_\mathcal{S})(\cdot)$. Intuitively, as visualized in Figure 1, if we can increase the margins between the class-specific modes in the source domain, domain shift will cause less performance degradation (Tommasi & Caputo, 2013).

We estimate the prototypical distribution in the embedding space as a parametric GMM as follows:

$$p_J(\boldsymbol{z}) = \sum_{j=1}^{k} \alpha_j \mathcal{N}(\boldsymbol{z}|\boldsymbol{\mu}_j, \boldsymbol{\Sigma}_j), \qquad (3)$$

where $\boldsymbol{\mu}_j$ and $\boldsymbol{\Sigma}_j$ denote the mean and co-variance matrices for each component and $\alpha_j$ denotes mixture weights for each component. We need to solve for these parameters to estimate the prototypical distribution. Note that unlike usual cases in which expectation maximization algorithm (Bilmes et al.,

1998) is used to estimate GMM parameters, the source data points are labeled. As a result, we can estimate $\boldsymbol{\mu}_j$ and $\boldsymbol{\Sigma}_j$ for each component independently using standard MAP estimates. Similarly, the weights $\alpha_j$ can be computed by a MAP estimate. Let $\boldsymbol{S}_j$ denote the support set for class $j$ in the training dataset, i.e., $\boldsymbol{S}_j = \{(\boldsymbol{x}_i^s, \boldsymbol{y}_i^s) \in \mathcal{D}_{\mathcal{S}} | \arg\max \boldsymbol{y}_i^s = j\}$. To cancel out outliers, we include only those source samples in the $\boldsymbol{S}_j$ sets, for which the source-trained model predicts the corresponding labels correctly. The MAP estimate for the mode parameters can be computed as:

$$\hat{\alpha}_j = \frac{|\boldsymbol{S}_j|}{N}, \quad \hat{\boldsymbol{\mu}}_j = \sum_{(\boldsymbol{x}_i^s, \boldsymbol{y}_i^s) \in \boldsymbol{S}_j} \frac{1}{|\boldsymbol{S}_j|} \phi_v(\boldsymbol{x}_i^s), \quad \hat{\boldsymbol{\Sigma}}_j = \sum_{(\boldsymbol{x}_i^s, \boldsymbol{y}_i^s) \in \boldsymbol{S}_j} \frac{1}{|\boldsymbol{S}_j|} \big(\phi_v(\boldsymbol{x}_i^s) - \hat{\boldsymbol{\mu}}_j\big)^\top \big(\phi_v(\boldsymbol{x}_i^s) - \hat{\boldsymbol{\mu}}_j\big).$$

(4)

Computations in Eq. equation 4 can be done efficiently. For a complexity analysis, please refer to the Appendix. Our idea is to use this prototypical distributional estimate to induce larger margins in the source domain (see Figure 1, right). We update the domain alignment term in equation 2 to induce larger margins. To this end, we update the source domain samples in the domain alignment term by samples of a labeled pseudo-dataset $\mathcal{D}_{\mathcal{P}} = (\mathbf{Z}_{\mathcal{P}}, \mathbf{Y}_{\mathcal{P}})$ that we generate using the GMM estimate, where $\boldsymbol{Z}_{\mathcal{P}} = [\boldsymbol{z}_1^p, \ldots, \boldsymbol{z}_{N_p}^p] \in \mathbb{R}^{p \times N_p}, \boldsymbol{Y}_{\mathcal{P}} = [\boldsymbol{y}_1^p, \ldots, \boldsymbol{y}_{N_p}^p] \in \mathbb{R}^{k \times N_p}$. This pseudo-dataset is generated using the prototypical distribution. We draw samples from the prototypical distributional estimate $\boldsymbol{z}_i^p \sim \hat{p}_J(\boldsymbol{z})$ for this purpose. To induce larger margins between classes, we feed the initial drawn samples into the classifier network and check the confidence level of the classifier about its predictions for these randomly drawn samples. We set a threshold $0 < \tau < 1$ level and select a subset of the drawn samples for which the confidence level of the classifier is more than $\tau$:

$$(\boldsymbol{z}_i^p, \boldsymbol{y}_i^p) \in \mathcal{D}_{\mathcal{P}} \quad \text{if} \quad \boldsymbol{z}_i^p \sim \hat{p}_J(\boldsymbol{z}) \quad \text{and} \quad \max\{h(\boldsymbol{z}_i^p)\} > \tau \quad \text{and} \quad \boldsymbol{y}_i^p = \arg\max_i\{h(\boldsymbol{z}_i^p)\}. \quad (5)$$

Given the GMM distributional form, selection of samples based on the threshold $\tau$ means that we include GMM samples that are closer to the class prototypes (see Figure 1). In other words, the margin between the clusters in the source domain increase if we use the generated pseudo-dataset for domain alignment. Hence, we update equation 2 and solve the following optimization problem:

$$\hat{\boldsymbol{v}}, \hat{\boldsymbol{w}} = \arg\min_{\boldsymbol{v}, \boldsymbol{w}} \frac{1}{N} \sum_{i=1}^{N} \mathcal{L}\big(h_{\boldsymbol{w}}(\phi_{\boldsymbol{w}}(\boldsymbol{x}_i^s)), \boldsymbol{y}_i^s\big) + \frac{1}{N_p} \sum_{i=1}^{N_p} \mathcal{L}\big(h_{\boldsymbol{w}}(\boldsymbol{z}_i^s), \boldsymbol{y}_i^s\big)$$
$$+ \lambda \hat{D}\big(\phi_{\boldsymbol{v}}(\boldsymbol{X}_{\mathcal{T}}), \boldsymbol{X}_{\mathcal{P}})\big) + \lambda \hat{D}\big(\phi_{\boldsymbol{v}}(\boldsymbol{X}_{\mathcal{S}}), \boldsymbol{X}_{\mathcal{P}}\big),$$

(6)

The first and the second terms in equation 6 are ERM terms for the source dataset and the generated pseudo-dataset to guarantee that the classifier continues to generalize well in the source domain after adaptation. The third and the fourth terms are empirical SWD losses (see Appendix for more details) that align the source and the target domain distributions using the pseudo-dataset which as we describe induces larger margins. The hope is that as visualized in Figure 1, these terms can reduce the effect of domain shift. Our proposed solution, named Sentiment Aanlysis Using Max-Margin classifiers (SAUM²), is presented and visualized in Algorithm 1 and Figure 1, respectively.

## 5 THEORETICAL ANALYSIS

We provide a theoretical justification for our algorithm. Following a standard PAC-learning framework, we prove that Algorithm 1 minimizes an upperbound for the target domain expected error. Consider that the hypothesis class in a PAC-learning setting is the family of classifier sub-networks $\mathcal{H} = \{h_{\boldsymbol{w}}(\cdot) | h_{\boldsymbol{w}}(\cdot) : \mathcal{Z} \to \mathbb{R}^k, \boldsymbol{v} \in \mathbb{R}^V\}$, where $V$ denotes the number of learnable parameters. We represent the expected error for a model $h_{\boldsymbol{w}}(\cdot) \in \mathcal{H}$ on the source and the target domains by $e_{\mathcal{S}}(\boldsymbol{w})$ and $e_{\mathcal{T}}(\boldsymbol{w})$, respectively. Given the source and the target datasets, we can represent the empirical source and target distributions in the embedding space as $\hat{\mu}_{\mathcal{S}} = \frac{1}{N} \sum_{n=1}^{N} \delta(\phi_{\boldsymbol{v}}(\boldsymbol{x}_n^s))$ and $\hat{\mu}_{\mathcal{T}} = \frac{1}{M} \sum_{m=1}^{M} \delta(\phi_{\boldsymbol{v}}(\boldsymbol{x}_m^t))$. Similarly, we can build an empirical distribution for prototypical distribution $\hat{\mu}_{\mathcal{P}} = \frac{1}{N_p} \sum_{q=1}^{N_p} \delta(\boldsymbol{z}_n^q)$. In our analysis we also use the notion of joint-optimal model $h_{\mathcal{S}, \mathcal{T}}(\cdot)$ in our analysis which is defined as: $\boldsymbol{w}^* = \arg\min_{\boldsymbol{w}} e_{\mathcal{S}, \mathcal{T}} = \arg\min_{\boldsymbol{w}}\{e_{\mathcal{S}} + e_{\mathcal{T}}\}$ for any given domains $\mathcal{S}$ and $\mathcal{T}$. When we have labeled data in both domains, this is the best model that can be trained using ERM. Existence of a good joint-trained model guarantees that the domains are related, e.g., similar sentiment polarities are encoded consistently across the two domains.

**Theorem 1**: Consider that we the procedure described in Algorithm 1 for cross-domain sentiment analysis, then the following inequality holds for the target expected error:

$$e_{\mathcal{T}} \leq e_{\mathcal{S}} + \hat{D}(\hat{\mu}_{\mathcal{S}}, \hat{\mu}_{\mathcal{P}}) + \hat{D}(\hat{\mu}_{\mathcal{T}}, \hat{\mu}_{\mathcal{P}}) + (1 - \tau) + e_{\mathcal{S},\mathcal{P}} + \sqrt{\left(2\log(\frac{1}{\xi})/\zeta\right)}\left(\sqrt{\frac{1}{N}} + \sqrt{\frac{1}{M}} + 2\sqrt{\frac{1}{N_p}}\right),$$
(7)

where $\xi$ is a constant number which depends on characteristic of the loss function $\mathcal{L}(\cdot)$.

**Proof:** The complete proof is included in the Appendix due to space limit.

Theorem 1 provides an explanation to justify Algorithm 1. We observe that all the terms in the upperbound of the target expected error in the right-hand side of equation 7 are minimized by $SAUM^2$ algorithm. The source expected error is minimized as the first term in equation 6. The second and the third terms terms are minimized as the third and fourth terms of equation 6. The fourth term $1 - \tau$ will be small if we set $\tau \approx 1$. Note however, when we select $\tau$ too close to 1, the samples will be centered around the prototypes. As a result, we will not match the higher-order distributional moments in the terms $\hat{D}(\hat{\mu}_{\mathcal{S}}, \hat{\mu}_{\mathcal{P}})$ and $\hat{D}(\hat{\mu}_{\mathcal{S}}, \hat{\mu}_{\mathcal{T}})$ and this can make the upperbound looser. The term $e_{\mathcal{S},\mathcal{P}}$ is minimized through the first and the second term of equation 6. This is highly important as using the pseudo-dataset provides a way to minimize this term. As can be seen in our proof in the Appendix, if we don't use the pseudo-dataset, this terms is replace with $e_{\mathcal{S},\mathcal{T}}$ which cannot be minimized directly due to lack of having annotated data in the target domain. The last term in equation 7 is a constant term that as common in PAC-learning can become negligible states that in order to train a good model if we have access to large enough datasets. Hence all the terms in the upperbound are minimized and if this upperbound is tight, then a good model will be trained for the target domain. If the two domain are related, e.g., share the same classes, and also classes become separable in the embedding space, i.e., GMM estimation error is small, then the upperbound is going to be likely tight. However, we highlight that possibility of a tight upperbound is a condition for our algorithm to work. This is a common limitation for most parametric algorithms.

## 6 EXPERIMENTAL VALIDATION

Our implemented code is available at `Appendix`.

### 6.1 EXPERIMENTAL SETUP

Most existing works report performance cross-domain tasks that are defined using the Amazon Reviews benchmark dataset (Blitzer et al., 2007). The dataset is built using Amazon product reviews from four product domains: Books (B), DVD (D), Electronics (E), and Kitchen (K) appliances. Each review is considered to have positive (higher than 3 stars) or negative (3 stars or lower) sentiment. Each Review is encoded in a 5000 dimensional or 30000 dimensional *tf-idf* feature vector of bag-of-words unigrams and bigrams. We report our performance on the 12 definable cross-domain tasks for this dataset. Each task consists of 2000 labeled reviews for the source domain and 2000 unlabeled reviews for the target domain, and 2500–5500 examples for testing. We report the average prediction accuracy and standard deviation (std) over 10 runs on the target domain testing split for our algorithm.

We compare our method against several recently developed algorithms. We compare against DSN (Bousmalis et al., 2016) CMD (Zellinger et al., 2017), ASYM (Saito et al., 2018), PBLM (Ziser & Reichart, 2018), MT-Tri (Ruder & Plank, 2018), TRL (Ziser & Reichart, 2019), and TAT (Liu et al., 2019). DSN and CMD are similar to $SAUM^2$ in that both align distributions in an embedding space. DSN learns shared and domain-specific knowledge for each domain and aligns the shared knowledge using the mean-based maximum mean discrepancy metric. CMD uses the central moment discrepancy metric for domain alignment. ASYM benefits from the idea of pseudo-labeling of the target samples to updated the base model. MT-Tri is based on ASYM but it also benefits from multi-task learning. TRL and PBLM do not use distribution alignment and are based on the pivot based language model. TAT is a recent work that has used adversarial learning successfully for cross-domain sentiment analysis. We provided results by the authors for the tasks in our table. We report std if std is reported in the original paper. All the methods except TAT that uses 30000 dimensional features use 5000 dimensional features. In our results, methods are comparable if they use features with the same dimension. We report performance of the source only (SO) model as a lowerbound.

| Task | B→D | B→E | B→K | D→B | D→E | D→K |
|---|---|---|---|---|---|---|
| SO | $81.7 \pm 0.2$ | $74.0 \pm 0.6$ | $76.4 \pm 1.0$ | $74.5 \pm 0.3$ | $75.6 \pm 0.7$ | $79.5 \pm 0.4$ |
| DSN | $82.8 \pm 0.4$ | $81.9 \pm 0.5$ | $84.4 \pm 0.6$ | $80.1 \pm 1.3$ | $81.4 \pm 1.1$ | $83.3 \pm 0.7$ |
| CMD | $82.6 \pm 0.3$ | $81.5 \pm 0.6$ | $84.4 \pm 0.3$ | $\mathbf{80.7} \pm 0.6$ | $82.2 \pm 0.5$ | $84.8 \pm 0.2$ |
| ASYM | 80.7 | 79.8 | 82.5 | 73.2 | 77.0 | 82.5 |
| PBLM | 84.2 | 77.6 | 82.5 | 82.5 | 79.6 | 83.2 |
| MT-Tri | 81.2 | 78.0 | 78.8 | 77.1 | 81.0 | 79.5 |
| TRL | 82.2 | - | 82.7 | - | - | - |
| $SAUM^2$ | $\mathbf{83.2} \pm 0.2$ | $\mathbf{83.9} \pm 0.3$ | $\mathbf{85.9} \pm 0.3$ | $80.3 \pm 0.4$ | $\mathbf{84.2} \pm 0.3$ | $\mathbf{87.3} \pm 0.2$ |
| TAT* | 84.5 | 80.1 | 83.6 | 81.9 | 81.9 | 84.0 |
| $SAUM^{2*}$ | $86.2 \pm 0.2$ | $85.1 \pm 0.2$ | $87.6 \pm 0.2$ | $80.9 \pm 0.5$ | $85.2 \pm 0.2$ | $88.6 \pm 0.2$ |

| Task | E→B | E→D | E→K | K→B | K→D | K→E |
|---|---|---|---|---|---|---|
| SO | $72.3 \pm 1.5$ | $74.2 \pm 0.6$ | $85.6 \pm 0.6$ | $73.1 \pm 0.1$ | $75.2 \pm 0.7$ | $85.4 \pm 1.0$ |
| DSN | $75.1 \pm 0.4$ | $77.1 \pm 0.3$ | $87.2 \pm 0.7$ | $76.4 \pm 0.5$ | $78.0 \pm 1.4$ | $86.7 \pm 0.7$ |
| CMD | $74.9 \pm 0.6$ | $77.4 \pm 0.3$ | $86.4 \pm 0.9$ | $75.8 \pm 0.3$ | $77.7 \pm 0.4$ | $86.7 \pm 0.6$ |
| ASYM | 73.2 | 72.9 | 86.9 | 72.5 | 74.9 | 84.6 |
| PBLM | 71.4 | 75.0 | 87.8 | 74.2 | $\mathbf{79.8}$ | $\mathbf{87.1}$ |
| MT-Tri | 73.5 | 75.4 | 87.2 | 73.8 | 77.8 | 86.0 |
| TRL | - | 75.8 | - | 72.1 | - | - |
| $SAUM^2$ | $\mathbf{78.6} \pm 0.4$ | $\mathbf{79.7} \pm 0.2$ | $\mathbf{89.2} \pm 0.2$ | $\mathbf{76.7} \pm 0.4$ | $79.1 \pm 0.4$ | $87.0 \pm 0.1$ |
| TAT* | 83.2 | 77.9 | 90.0 | 75.8 | 77.7 | 88.2 |
| $SAUM^{2*}$ | $78.8 \pm 0.3$ | $78.9 \pm 0.3$ | $90.1 \pm 0.2$ | $78.1 \pm 0.2$ | $78.8 \pm 0.4$ | $88.1 \pm 0.1$ |

Table 1: Classification accuracy for the cross-domain sentiment analysis tasks for Amazon Reviews dataset. In this table, * denotes methods that use 30000 dimensional *tf-idf* feature vectors.

We used the benchmark network architecture that is used in the above mentioned works. We used an encoder with one hidden dense layer with 50 nodes with sigmoid activation function. The classifiers consist of a softmax layer with two output nodes. We implemented our method in Keras, used adam optimizer, and tuned the learning rate in the source domain. We set $\tau = 0.99$ and $\lambda = 10^{-2}$. We observed empirically that our algorithm is not sensitive to the value of $\lambda$.

## 6.2 RESULTS

Our results are reported in Table 1. In this table, bold font denotes best performance among the methods that use 5000 dimensional features. We see that $SAUM^2$ algorithm performs reasonably well and in most cases leads to the best performance. Note that this is not unexpected as none of the methods has the best performance across all tasks. We observe from this table that overall the methods DSN and CMD which are based on aligning the source and target distributions- which are more similar to our algorithm- have relatively similar performances. This observation suggests that we should not expect considerable performance boost if we simply align the distributions by designing a new alignment loss function. This means that outperformance of $SAUM^2$ compared to these methods likely stems from inducing larger margins. We verify this intuition in our ablative study. We also observe that increasing the dimensional of *tf-idf* features to 30000 leads to performance boost which is probably the reason behind good performance of TAT compared to the rest of methods. Hence, we need to use the same dimension for features for fair comparison among the methods.

To provide an intuition for the rationale we used, we have used UMAP McInnes et al. (2018) visualization tool to reduce the dimension of the data representations in the 50D embedding space to two for the purpose of 2D visualization. Figure 2 visualizes the testing splits of the source domain before model adaptation, the testing splits of the target domain before and after model adaptation, and finally random samples drawn from the prototypical distribution for the D→K task. Each point represents one data point and each color represents one of the sentiments. Observing Figure 2a and Figure 2b, we conclude that GMM prototypical distribution approximates the source domain distribution reasonably well and at the same time, a margin between the classes in the boundary region is observable. Figure 2c visualizes the target domain samples prior to model adaptation. As expected, we observe that domain gap has caused less separations between the classes, as also evident from SO performance in Table 1. Figure 2d visualizes the target domain samples after adaptation using $SAUM^2$ algorithm. Comparing Figure 2d with Figure 2c and Figure 2a, we see that the classes

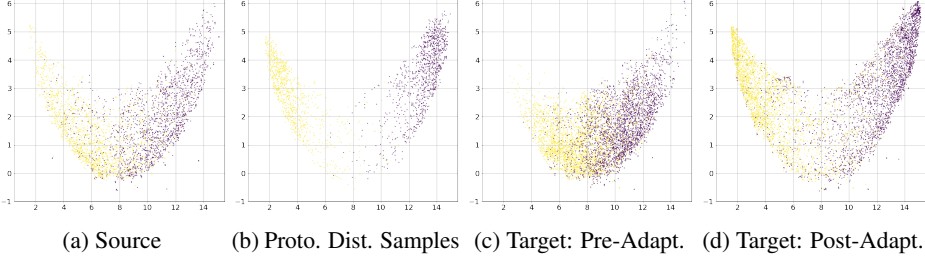

(a) Source     (b) Proto. Dist. Samples    (c) Target: Pre-Adapt.    (d) Target: Post-Adapt.

Figure 2: UMAP visualization for the task D→K task: (a) the source domain testing split, (b) the prototypical distribution samples, (c) the target domain testing split prior to adaptation, and (d) the target domain testing split after adaptation. (Best viewed in color).

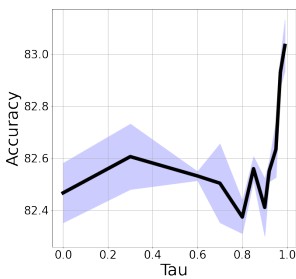

Figure 3: Effect of $\tau$ on performance.

Table 2: Performance of AO setting.

| Task | B→D | B→E | B→K |
|------|------|------|------|
| | $81.9 \pm 0.5$ | $80.9 \pm 0.8$ | $83.2 \pm 0.8$ |
| Task | D→B | D→E | D→K |
| | $74.0 \pm 0.9$ | $80.9 \pm 0.7$ | $83.4 \pm 0.6$ |
| Task | E→B | E→D | E→K |
| | $74.1 \pm 0.6$ | $74.0 \pm 0.3$ | $87.8 \pm 0.9$ |
| Task | D→B | D→E | D→K |
| | $74.1 \pm 0.6$ | $74.0 \pm 0.3$ | $87.8 \pm 0.9$ |

have become more separated. Also, careful comparison of Figure 2d and Figure 2b reveals $SAUM^2$ algorithm has led to a bias in the target domain to move the data points further from the boundary.

### 6.3 ABLATION STUDIES

First note that the source only (SO) model result, which is trained using equation 1, already serve as a basic ablative study to verify the effect of domain alignment. Improvement over this baseline demonstrates effect of using the information which is encoded in the target unlabeled data.

In Table 2, we have provided an additional ablative studies. We have reported result of alignment only (AO) model adaptation based on equation 2. The AO model does not benefit from the margins that $SAUM^2$ algorithm induces between the classes. Comparing AO results with Table 1, we can conclude that the effect of increased margins is important in our performance. Compared to other cross-domain sentiment analysis methods, the performance boost for our algorithm stems from inducing large margins. This suggests that researchers may check to investigate secondary techniques for domain adaptation in NLP domains, in addition to probability distribution alignment.

Finally, we have studied the effect of the value of the confidence parameter on performance. In Figure 3, we have visualized the performance of our algorithm for the task $B \rightarrow D$ when $\tau$ is varied in the interval $[0, 0.99]$. When $\tau = 0$, the samples are not necessarily confident samples. We observe that as we increase the value of $\tau$, the performance increases as a result of inducing larger margins. For values $\tau > 0.8$, the performance has less variance which suggests robustness of performance if $\tau \approx 1$. These empirical observations about $\tau$ accord with our theoretical result, stated in equation 7.

## 7 CONCLUSIONS

We developed a method for cross-domain sentiment analysis based on aligning two domain-specific distributions in a shared embedding space. We demonstrated that one can improve upon this baseline by inducing larger margins between the classes in the source domain using an intermediate multi-modal prototypical distribution. As a result, the domain shift problem is mitigated in the target domain. Our experiments demonstrate that our algorithm is effective. A future research direction is to address cross-domain sentiment analysis when different types of sentiment exists across the domains.

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

# A APPENDIX

## A.1 SLICED WASSERSTEIN DISTANCE

We relied on minimizing the Sliced Wasserstein (SWD) distance for domain alignment. SWD is defined based on the Wasserstein distance (WD) and is a mean to come up with a more computationally efficient distribution metric. The WD between two distributions $p_{\mathcal{S}}$ and $p_{\mathcal{T}}$, is defined as:

$$W_c(p_{\mathcal{S}}, p_{\mathcal{T}}) = \inf_{\gamma \in \Gamma(p_{\mathcal{S}}, p_{\mathcal{T}})} \int_{X \times Y} c(x, y) d\gamma(x, y) \qquad (8)$$

where $\Gamma(p_{\mathcal{S}}, p_{\mathcal{T}})$ is the set of all joint distributions $p_{\mathcal{S}, \mathcal{T}}$ with marginal single variable distributions $p_{\mathcal{S}}$ and $p_{\mathcal{T}}$, and $c : X \times Y \to \mathbb{R}^+$ is the cost function, e.g., $\ell_2$-norm Euclidean distance.

We observe that computing WD involves solving a complicated optimization problem in the general case. However, when the two distributions are $1-$dimensional, WD has a closed-form solution:

$$W_c(p_{\mathcal{S}}, p_{\mathcal{T}}) = \int_0^1 c(P_{\mathcal{S}}^{-1}(\tau), P_{\mathcal{T}}^{-1}(\tau)) d\tau, \qquad (9)$$

where $P_{\mathcal{S}}$ and $P_{\mathcal{T}}$ are the cumulative distributions of the 1D distributions $p_{\mathcal{S}}$ and $p_{\mathcal{T}}$. This closed-form solution motivates the definition of SWD in order to extend applicability of equation 9 for higher dimensional distributions.

SWD is defined based on the idea of slice sampling Neal (2003). The idea is to project two $d$-dimensional distributions into their marginal one-dimensional distributions along a subspace, i.e., slicing the high-dimensional distributions, and then compute the distance between the two distribution by integrating over all the WD between the resulting 1D marginal probability distributions over all possible 1D subspaces using the closed form solution of WD. This can be a good replacement for the WD as any probability distribution can be expressed by the set of $1-$dimensional marginal projection distributions Helgason (2011). More specifically, a one-dimensional slice of the distribution for the distribution $p_{\mathcal{S}}$ is defined:

$$\mathcal{R}p_{\mathcal{S}}(t; \boldsymbol{\gamma}) = \int_{\mathcal{S}^{d-1}} p_{\mathcal{S}}(\boldsymbol{x}) \boldsymbol{\delta}(t - \langle \boldsymbol{\gamma}, \boldsymbol{x} \rangle) d\boldsymbol{x}, \qquad (10)$$

where $\boldsymbol{\delta}(\cdot)$ denotes the Kronecker delta function, $\langle \cdot, \cdot \rangle$ denotes the vector inner dot product, $\mathbb{S}^{d-1}$ is the $d$-dimensional unit sphere, and $\boldsymbol{\gamma}$ is the projection direction.

The SWD is defined as the integral of all WD between the sliced distributions over all 1D subspaces $\boldsymbol{\gamma}$ on the unit sphere as follows:

$$SW(p_{\mathcal{S}}, p_{\mathcal{T}}) = \int_{\mathbb{S}^{d-1}} W(\mathcal{R}p_{\mathcal{S}}(\cdot; \gamma), \mathcal{R}p_{\mathcal{T}}(\cdot; \gamma)) d\gamma \qquad (11)$$

The main advantage of using SWD is that, computing SWD does not require solving a numerically expensive optimization.

In our practical setting, only samples from the distributions are available and we don't have the distributional form. Another advantage of SWD is that its empirical version can be computed based on the one-dimensional empirical WD. One-dimensional empirical WD be approximated as the $\ell_p$-distance between the sorted samples. We can compute merely the integrand function in equation 11 for a known $\gamma$ and then the integral in equation 11 via Monte Carlo style numerical integration. To this end, we draw random projection subspace $\boldsymbol{\gamma}$ from a uniform distribution that is defined over the unit sphere and then compute 1D WD along this sample. We can then approximate the integral in equation 11 by computing the arithmetic average over a suitably large enough number of drawn samples. More specifically, the SWD between $f$-dimensional samples $\{\phi(\mathbf{x}_i^{\mathcal{S}}) \in \mathbb{R}^f \sim p_{\mathcal{S}}\}_{i=1}^M$ and $\{\phi(\mathbf{x}_i^{\mathcal{T}}) \in \mathbb{R}^f \sim p_{\mathcal{T}}\}_{j=1}^M$ in our setting can be approximated as the following sum:

$$SW^2(p_{\mathcal{S}}, p_{\mathcal{T}}) \approx \frac{1}{L} \sum_{l=1}^L \sum_{i=1}^M |\langle \gamma_l, \phi(\mathbf{x}_{s_l[i]}^{\mathcal{S}}) \rangle - \langle \gamma_l, \phi(\mathbf{x}_{t_l[i]}^{\mathcal{T}}) \rangle|^2 \qquad (12)$$

where $\gamma_l \in \mathbb{S}^{f-1}$ is uniformly drawn random sample from the unit $f$-dimensional ball $\mathbb{S}^{f-1}$, and $s_l[i]$ and $t_l[i]$ are the sorted indices of $\{\gamma_l \cdot \phi(\mathbf{x}_i)\}_{i=1}^M$ for source and target domains, respectively.

We utilize this empirical version of SWD in equation 12 to align the distributions in the embedding space. Note that the function in equation 12 is differentiable with respect to the encoder parameters and hence we can use gradient-based optimization techniques to minimize it with respect to the model parameters.

## A.2 PROOF OF THEOREM 1

We use the following theorem by Redko et al. Redko & Sebban (2017) and a result by Bolley Bolley et al. (2007) on convergence of the empirical distribution to the true distribution in terms of the WD distance in our proof.

**Theorem 2 (Redko et al. Redko & Sebban (2017)):** Under the assumptions described in our framework, assume that a model is trained on the source domain, then for any $d' > d$ and $\zeta < \sqrt{2}$, there exists a constant number $N_0$ depending on $d'$ such that for any $\xi > 0$ and $\min(N, M) \geq \max(\xi^{-(d'+2),1})$ with probability at least $1 - \xi$, the following holds:

$$e_{\mathcal{T}} \leq e_{\mathcal{S}} + W(\hat{\mu}_{\mathcal{T}}, \hat{\mu}_{\mathcal{S}}) + e_{\mathcal{S},\mathcal{T}} + \sqrt{\left(2\log(\frac{1}{\xi})/\zeta\right)}\left(\sqrt{\frac{1}{N}} + \sqrt{\frac{1}{M}}\right). \tag{13}$$

Theorem 2 provides an upperbound for the performance of a source-trained model in the target domain Redko et al. Redko & Sebban (2017) prove Theorem 2 for a binary classification setting. We also provide our proof in this case but it can be extended.

The second term in Eq. equation 13 demonstrates the effect of domain shift on the performance of a source-trained model in a target domain. When the distance between the two distributions is significant, this term will be large and hence the upperbound in Eq. equation 13 will be loose which means potential performance degradation. Our algorithm mitigates domain gap because this term is minimized by minimization of the second and the third terms in Theorem 1.

**Theorem 1** : Consider that we the procedure described in Algorithm 1 for cross-domain sentiment analysis, then the following inequality holds for the target expected error:

$$e_{\mathcal{T}} \leq e_{\mathcal{S}} + \hat{D}(\hat{\mu}_{\mathcal{S}}, \hat{\mu}_{\mathcal{P}}) + \hat{D}(\hat{\mu}_{\mathcal{T}}, \hat{\mu}_{\mathcal{P}}) + (1-\tau) + e_{\mathcal{S},\mathcal{P}} + \sqrt{\left(2\log(\frac{1}{\xi})/\zeta\right)}\left(\sqrt{\frac{1}{N}} + \sqrt{\frac{1}{M}} + 2\sqrt{\frac{1}{N_p}}\right), \tag{14}$$

where $\xi$ is a constant which depends on $\mathcal{L}(\cdot)$ and $e'_C(\boldsymbol{w}^*)$ denotes the expected risk of the optimally joint trained model when used on both the source domain and the pseudo-dataset.

**Proof:** Due to the construction of the pseudo-dataset, the probability that the predicted labels for the pseudo-data points to be false is equal to $1 - \tau$. Let:

$$|\mathcal{L}(h_{\boldsymbol{w}_0}(\boldsymbol{z}_i^p), \boldsymbol{y}_i^p) - \mathcal{L}(h_{\boldsymbol{w}_0}(\boldsymbol{z}_i^p), \hat{\boldsymbol{y}}_i^p)| = \begin{cases} 0, & \text{if } \boldsymbol{y}_i^t = \hat{\boldsymbol{y}}_i^t. \\ 1, & \text{otherwise.} \end{cases} \tag{15}$$

We use Jensen's inequality and take expectation on both sides of equation 15 to deduce:

$$|e_{\mathcal{P}} - e_{\mathcal{T}}| \leq \mathbb{E}\left(|\mathcal{L}(h_{\boldsymbol{w}_0}(\boldsymbol{z}_i^p), \boldsymbol{y}_i^p) - \mathcal{L}(h_{\boldsymbol{w}_0}(\boldsymbol{z}_i^p), \hat{\boldsymbol{y}}_i^p)|\right) \leq (1-\tau). \tag{16}$$

Applying equation 16 in the below, deduce:

$$e_{\mathcal{S}} + e_{\mathcal{T}} = e_{\mathcal{S}} + e_{\mathcal{T}} + e_{\mathcal{P}} - e_{\mathcal{P}} \leq e_{\mathcal{S}} + e_{\mathcal{P}} + |e_{\mathcal{T}} - e_{\mathcal{P}}| \leq e_{\mathcal{S}} + e_{\mathcal{P}} + (1-\tau). \tag{17}$$

Taking infimum on both sides of equation 17, we deduce:

$$e_{\mathcal{S},\mathcal{T}} \leq e_{\mathcal{S},\mathcal{P}} + (1-\tau). \tag{18}$$

Now by considering Theorem 2 for the two domains $\mathcal{S}$ and $\mathcal{T}$ and then using equation 18 in equation 13, we can conclude:

$$e_{\mathcal{T}} \leq e_{\mathcal{S}} + D(\hat{\mu}_{\mathcal{T}}, \hat{\mu}_{\mathcal{S}}) + e_{\mathcal{S},\mathcal{P}} + (1-\tau) + \sqrt{\left(2\log(\frac{1}{\xi})/\zeta\right)}\left(\sqrt{\frac{1}{N}} + \sqrt{\frac{1}{M}}\right). \tag{19}$$

Now using the triangular inequality on the metrics we can deduce:

$$D(\hat{\mu}_{\mathcal{T}}, \hat{\mu}_{\mathcal{S}}) \leq D(\hat{\mu}_{\mathcal{T}}, \mu_{\mathcal{P}}) + D(\hat{\mu}_{\mathcal{S}}, \mu_{\mathcal{P}}) \leq D(\hat{\mu}_{\mathcal{T}}, \hat{\mu}_{\mathcal{P}}) + D(\hat{\mu}_{\mathcal{S}}, \hat{\mu}_{\mathcal{P}}) + 2D(\hat{\mu}_{\mathcal{P}}, \mu_{\mathcal{P}}). \quad (20)$$

Now we replace the term $D(\hat{\mu}_{\mathcal{P}}, \mu_{\mathcal{P}})$ with its empirical counterpart using Theorem 1.1 in the work by Bolley et al. (2007).

**Theorem 3** (Theorem 1.1 by Bolley et al. Bolley et al. (2007)): consider that $p(\cdot) \in \mathcal{P}(\mathcal{Z})$ and $\int_{\mathcal{Z}} \exp(\alpha \|\boldsymbol{x}\|_2^2) dp(\boldsymbol{x}) < \infty$ for some $\alpha > 0$. Let $\hat{p}(\boldsymbol{x}) = \frac{1}{N} \sum_i \delta(\boldsymbol{x}_i)$ denote the empirical distribution that is built from the samples $\{\boldsymbol{x}_i\}_{i=1}^{N}$ that are drawn i.i.d from $\boldsymbol{x}_i \sim p(\boldsymbol{x})$. Then for any $d' > d$ and $\xi < \sqrt{2}$, there exists $N_0$ such that for any $\epsilon > 0$ and $N \geq N_o \max(1, \epsilon^{-(d'+2)})$, we have:

$$P(W(p, \hat{p}) > \epsilon) \leq \exp(-\frac{-\xi}{2} N \epsilon^2), \quad (21)$$

where $W$ denotes the WD distance. This relation measures the distance between the empirical distribution and the true distribution, expressed in the WD distance.

Applying equation 20 and equation 21 on equation 19 concludes Theorem 2 as stated:

$$e_{\mathcal{T}} \leq e_{\mathcal{S}} + D(\hat{\mu}_{\mathcal{S}}, \hat{\mu}_{\mathcal{P}}) + D(\hat{\mu}_{\mathcal{T}}, \hat{\mu}_{\mathcal{P}}) + (1 - \tau) + e_{\mathcal{S},\mathcal{P}} + \sqrt{(2\log(\frac{1}{\xi})/\zeta)} \left( \sqrt{\frac{1}{N}} + \sqrt{\frac{1}{M}} + 2\sqrt{\frac{1}{N_p}} \right), \quad (22)$$

### A.3 COMPLEXITY ANALYSIS FOR GMM ESTIMATION

Estimating a GMM distribution usually is a computationally expensive tasks. The major reason is that normally the data points are unlabeled. This would necessitate relying on iterative algorithms such expectation maximization (EM) algorithm Moon (1996). Preforming iterative E and M steps until convergence leads to high computational complexity Roweis (1998). However, estimating the prototypical distribution with a GMM distribution is much simpler in our learning setting. Existence of labels helps us to decouple the Gaussian components and compute the parameters using MAP estimate for each of the mode parameters in one step as follows:

$$\hat{\alpha}_j = \frac{|\boldsymbol{S}_j|}{N}, \quad \hat{\boldsymbol{\mu}}_j = \sum_{(\boldsymbol{x}_i^s, \boldsymbol{y}_i^s) \in \boldsymbol{S}_j} \frac{1}{|\boldsymbol{S}_j|} \phi_v(\boldsymbol{x}_i^s), \quad \hat{\boldsymbol{\Sigma}}_j = \sum_{(\boldsymbol{x}_i^s, \boldsymbol{y}_i^s) \in \boldsymbol{S}_j} \frac{1}{|\boldsymbol{S}_j|} \left( \phi_v(\boldsymbol{x}_i^s) - \hat{\boldsymbol{\mu}}_j \right)^{\top} \left( \phi_v(\boldsymbol{x}_i^s) - \hat{\boldsymbol{\mu}}_j \right). \quad (23)$$

Given the above and considering that the source domain data is balanced, complexity of computing $\alpha_j$ is $O(N)$ (just checking whether data points $\boldsymbol{x}_i^s$ belong to $\boldsymbol{S}_j$). Complexity of computing $\boldsymbol{\mu}_j$ is $O(NF/k)$, where $F$ is the dimension of the embedding space. Complexity of computing the covariance matrices $\boldsymbol{\Sigma}_j$ is $O(F(\frac{N}{k})^2)$. Since, we have $k$ components, the total complexity of computing GMM is $O(\frac{FN^2}{k})$. If $O(F) \approx O(k)$, which seems to be a reasonable practical assumption, then the total complexity of computing GMM would be $O(N^2)$. Given the large number of learnable parameters in most deep neural networks which are more than $N$ for most cases, this complexity is fully dominated by complexity of a single step of backpropagation. Hence, this computing the GMM parameters does not increase the computational complexity for.

### A.4 ADDITIONAL EXPERIMENTS ON IMBALANCED DATA

We have used a balanced dataset in terms of class labels in our experiments. However, the label distribution for the target domain training dataset cannot be enforced to be balanced in practical applications due to absence of labels. This a highly unexplored challenge in the domain adaptation literature which we study in this section. To study the effect of label imbalance using a controlled experiment, we synthetically design an imbalanced dataset using the Amazon dataset. We repeat the experiments we had, only with the difference of using imbalanced target domain datasets. We design two experiments by only include 111 and 250 data points from class one such that the target domain datasets has the 90/10 and 80/20 ratios of imbalance between the two classes, respectively. We have provided domain adaptation results using for these two imbalanced scenarios in Table 3. We can see

| Task | B→D | B→E | B→K | D→B | D→E | D→K |
|------|-----|-----|-----|-----|-----|-----|
| 80/20 | $82.8 \pm 0.3$ | $83.2 \pm 0.5$ | $85.5 \pm 0.3$ | $78.7 \pm 0.2$ | $83.3 \pm 0.2$ | $86.8 \pm 0.2$ |
| 90/10 | $82.9 \pm 0.5$ | $83.4 \pm 0.3$ | $85.8 \pm 0.2$ | $78.5 \pm 0.4$ | $83.3 \pm 0.4$ | $86.8 \pm 0.3$ |

| Task | E→B | E→D | E→K | K→B | K→D | K→E |
|------|-----|-----|-----|-----|-----|-----|
| 80/20 | $78.7 \pm 0.2$ | $78.5 \pm 0.5$ | $88.6 \pm 0.1$ | $76.3 \pm 0.6$ | $77.9 \pm 0.4$ | $86.6 \pm 0.1$ |
| 90/10 | $78.7 \pm 0.2$ | $78.0 \pm 0.4$ | $88.0 \pm 0.2$ | $76.5 \pm 0.5$ | $77.3 \pm 0.3$ | $86.7 \pm 0.2$ |

Table 3: Effect of label imbalance in the target domain on the proposed method.

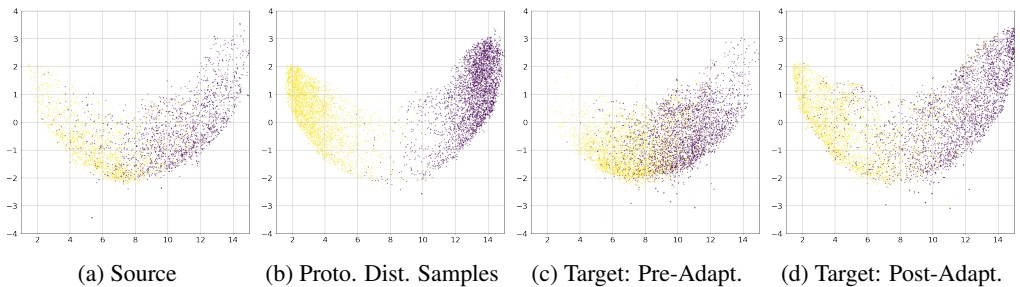

(a) Source     (b) Proto. Dist. Samples     (c) Target: Pre-Adapt.     (d) Target: Post-Adapt.

Figure 4: UMAP visualization for the task D→K task in the imbalanced regime of $90/10$: (a) the source domain testing split, (b) the prototypical distribution samples, (c) the target domain testing split prior to adaptation, and (d) the target domain testing split after adaptation. (Best viewed in color).

that performance of our algorithm slightly has degraded but our algorithm has been robust to a large extent with respect to label imbalance.

As a secondary sanity check, we have presented the UMAP visualization for the testing data split of the task D→K for the imbalanced $90/10$ task in Figure 4. Observations in Figure 4 match what we reported in Table 3, confirming that our algorithm does suffer considerable degradation if the target domain data imbalanced for the tasks built using the Amazon reviews dataset.

