# OpenReview forum: "Learning a Max-Margin Classifier for Cross-Domain Sentiment Analysis"
_ICLR.cc/2021/Conference — Reject_

### Official Review · AnonReviewer4 · 2020-10-26
**Learning a Max-Margin Classifier for Cross-Domain Sentiment Analysis**

**Rating:** 5
**Confidence:** 4

**Review:**

Pros:
The proposed SAUM2 is novel. As a contribution, the work introduced large margins between classes in a source domain. This is relevant in a cross-domain sentiment analysis problem. The result shows that the resultant model is domain-independent which is Ok for a general application.

Cons:
Although the SAUM2 is novel in my opinion, the algorithm seems cumbersome. It is unclear how the "domain shift" would cause performance degradation. This is so because it was not shown how Theorem 1 was an explanation of Algorithm 1. The idea looks good in my assessment but it was not shown how it would be reproducible in subsequent works.

The following minor issues need to be revised:
1. Abstract: The 3rd sentence needs to be revised - 'per each domain' is ambiguous.
2. Introduction: Line 9 - 'in a other different domains' needs to be revised. In the second paragraph, the term 'domain-agnostic' needs to be defined or explained. On page 2, 'SWD' needs to be defined also or refer to its definition in Pg4, Para 1. Also on Page 2, under contributions, the phrase 'in the embedding matches this distribution' is ambiguous.
3. Page 6, Para. 1: The section of the Appendix referred to should be specified.
4. Page 8, Section 6.3: 'we may not matching the higher order' is ambiguous.

---

> ### Author Response · Authors · 2020-11-17
> **Answer to Reviewer 4**
>
> Thank you for your positive assessment and for reading the paper carefully. We responded to your points below.
>
> 1. Part of what you stated is because our proof is included in the Appendix. We added further explanations in the proof in the Appendix, after Theorem 2,  to make better connections. We hope this addresses your concern.
>
> 2. Thank you for pointing out the typos. We corrected them. We will make sure to have the paper proofread by a professional editor in our organization for the camera-ready version.

---

### Official Review · AnonReviewer3 · 2020-10-26
**Interesting paper, but there are a few issues**

**Rating:** 5
**Confidence:** 4

**Review:**

Summary:
The paper proposes an unsupervised domain adaptation approach for sentiment analysis. The main idea is to align the cross-domain representation with Sliced Wasserstein Distance (SWD), and train the sentiment classifier on additional pseudo-labeled target domain data. By modeling the embedding spaces as a mixture of Gaussians (GMM), with one Gaussian per sentiment class, the authors sample for pseudo-labeled data with the GMM and retain only the samples that are labeled confidently by the classifier. The authors presented a theoretical analysis leveraging on previous results for SWD for domain adaptation.

Strengths:
- the paper presents an appealing representation learning approach to cross-domain alignment of representations.

Weaknesses:
- By only retaining samples with high classifier confidence, the authors claim that "the margin between the clusters in the source domain increase if we use the generated pseudo-dataset for domain alignment". This seems to be an important claim in this paper, as the authors named their approach a "max-margin" approach, and they argue that it is this effect that is the reason for the good performance of their model. However, there is no proof in the paper of this claim, and I am not sure if that is true in a mathematical sense.

- As a general domain adaptation paper, the results would be more convincing with another data set. As an application paper for cross-domain sentiment analysis, the authors failed to compare against recent work such as [2], [3] and [4]. Du et al. [2] used the BERT representation, but in Table 1 in [2], the results of IATN [3] and HATN [4] seemed to outperform this current submission without using BERT.

Minor comments:
- Wrong reference for TAT, should be [1]
- Color in Figure 2 could be more distinct.
- K->E: PBLM has a higher score (87.1) but SAUM^2's 86.8 was bolded
- The paper did not explain how the standard deviations in Table 1 are obtained, for rows that have them.
- The authors should proof read the paper for typos and grammatical mistakes, e.g., "costumers", "Due to existence" missing "the", "we use the common of reducing", "our goal is mitigate" missing "to", "classier",


[1] Hong Liu, Mingsheng Long, Jianmin Wang, and Michael Jordan. Transferable adversarial training: A general approach to adapting deep classifiers. ICML 2019.
[2] Chunning Du, Haifeng Sun, Jingyu Wang, Qi Qi, Jianxin Liao. Adversarial and Domain-Aware BERT for Cross-Domain Sentiment Analysis. ACL 2020.
[3] Kai Zhang, Hefu Zhang, Qi Liu, Hongke Zhao, Hengshu Zhu, Enhong Chen. Interactive Attention Transfer Network for Cross-Domain Sentiment Classiﬁcation. AAAI 2019
[4] Zheng Li, Ying Wei, Yu Zhang, Qiang Yang. Hierarchical Attention Transfer Network for Cross-Domain Sentiment Classiﬁcation. AAAI 2018

Comments after authors' rebuttal:
Thanks for addressing my comments. However, I think the current submission needs further work.
- The authors agreed with me on my point that the "max-margin" claim might need further work, and since this is an important claim in the paper, I cannot improve my review score after the author response.
- Different data splits should not be a barrier for comparison against previous work.

---

> ### Author Response · Authors · 2020-11-17
> **Answer to Reviewer 3**
>
> Thank you for your feedback. It seems that despite finding our work appealing, you have concerns about both the algorithm and also the results. We have responded to your comments below. We think addressing your comments about the experiments is more straightforward and we hope you reconsider your rating.
>
> 1. Hopefully, we can state our reasoning more clearly here. Please refer to Figure 1 and note the visualization for embedding space on both sides of the figure. We have tried to visualize the typical data representation in the embedding space in the two panels of Figure 1. If we can model the source distribution with a GMM quite well, as in the left-hand side of Figure 1, and then choose high-confidence samples of GMM, we will have a situation similar to the right-hand side of Figure 1. We meant that if we use the high-confident samples and solve Eq 6, data representations for source and target domain would be moved towards the high confidence region and as a result, the margin between the classes will increase similar to the right-hand side of Figure 1. This intuition has been empirically verified in Figure 2 (a), (b), and (d) which reiterate Figure 1 with real data. Judging from these subfigures, you can see that the margin between the classes has increased as a result of using our algorithm. However, we agree with your point about the term “max-margin” in the title. We are open to changing the title according to your suggestion because the margin is not the maximum possible margin in our method and hopefully other researchers are going to develop UDA methods that outperform our method in terms of inducing larger margins.
>
> 2. Thank you for pointing out these works. Please note that:
>
> HATN : their experimental setup is different from the common setting in the literature and hence comparing against them will not be fair. First, they are using 5000 data points for training which is significantly larger than the common value of 2000 data points in the literature. All the papers we have listed use 2000 training data points. Second, they use word2vec to embed the reviews. For these reasons, we don’t think it is fair to add HATN to our table. The good performance of HATN likely is because of using more training data, rather than success in domain adaptation.
>
> IATN:  The same is true about HATN. They use even more training data points, i.e, 5600 data points for training, and also use word2vec. Using more training data is likely a partial reason why IATN outperforms HATN.
>
> In conclusion, it is not fair to compare HATN and IATN with our method and other methods in our tables. Of course, choosing the testing/training split is somewhat arbitrary and there is nothing wrong with what HATN and IATN works have done. But we followed what is more common in the literature. The same logic works for Du et al because they use BERT features which means they use an additional large dataset that has been used to train BERT. Also, we noted that the results Du et al report for IATN are exactly copied from the IATN original paper and hence the above logic is correct. Their reported results for HATN are slightly different from the original paper but it is not clear why this disparity exists in the Du et al paper.
>
> As a sanity check, we performed one experiment by using 5600 data points for training and 400 for testing, and the performance on B->E task boosted to 86.3 which is comparable with 85.7 of IATN and 86.5 of HATN. Note that adding word2vec would likely give us a further boost. For the above reasoning, we did include the above works in our tables.
>
> 3. Thank you for pointing out the typos. We corrected them in the paper. We will make sure to have the paper proofread by a professional editor in our organization for the camera-ready version.

---

> ### Author Response · Authors · 2020-11-24
> **Max-margin claim**
>
> Thank you for reading our rebuttal and updating your response. We wanted to add:
>
> 1. We agree with you that the term "max-margin" is not accurate and we could be more careful. Now our question is: is your concern only about using this term? If yes, wouldn't changing the term address your concern? For example, if we use the term "increased-margin" instead, will your concern be addressed? What "further work" can be done to address your concern in this direction?
>
> 2. We would like to note that reporting our results using a "different data split" is definitely possible and there is no barrier to compare against IATN and HATN. We can redo our experiments using the data splits that each of those papers has used. This will require changing 2-3 lines of our code and then rerun the code. We are quite certain that our results will boost as a result of using more training data. This could also be seen in the task that we included in our initial rebuttal. By no means we wanted to discard your comment, but as we pointed out, the splits used by IATN and HATN are different from what most works have used to report their performance. Hence, comparing other papers against IATN and HATN is unfair.  Since we will have an additional page for the camera-ready version, we can perform experiments based on splits used in IATN and HATN and add them in separate tables in the paper. If by "further work" you mean adding these experiments, we are very certain we can add them to the paper for the camera-ready version. Please clarify that this will address your concern and then we can add those additional results.
>
> Best,
> Our team

---

### Official Review · AnonReviewer2 · 2020-10-27

**Rating:** 5
**Confidence:** 4

**Review:**

In this paper, the authors propose a new method to improve cross-domain sentiment analysis. They introduce large margins between classes in the source domain and it helps to reduce the possibility of misclassification on the target domain because of domain drift. After the training on the source domain, they use a Gaussian mixture model to generalize the samples and then select a subset of the drawn samples from the learned GMM for which the confidence level of the classifier is more than a predefined threshold. Therefore the drawn samples will keep larger margin between classes in the source domain. The authors provide a theoretic analysis to justify the algorithm. The experimental results demonstrate the efficacy of their method.


Pros:
1. The authors enlarge the margin between classes on the source domain to reduce the possibility of missclassification on the target domain because of domain drift.
2. The paper analyzes their method in a theoretic way.

Cons:
1. The proposed method is very straightforward. The 'max-margin' component in the paper is implemented by the selection of samples with higher confidence of the classifier. Therefore, the performance depends on careful tuning of the threshold. A higher threshold will result in lots of samples in the source domain will be filtered out and then there is information loss during the alignment between source domain and target domain.

2. The evaluation is weak. Some experimental settings are unreasonable.

    a) The baseline methods use different dimensions of feature vectors. Some use 5000 and the other use 30000 as the vector size. For the fair comparison, both settings of 5000 and 30000 should be considered for the baseline methods. Perhaps some methods are sensitive to the variance of the vector size.

    b) How to set the threshold \tau? The author uses 0.95 without any explanation.

    c) It is not enough that only two values, i.e. 0 and 0.99 are assigned in Table2. A line should be drawn to depict the variance of performance according to different values of the threshold.

Typos:
There are some typos in the paper. The authors should proofread the draft carefully. I list some typos as follows:

'costumers' --> 'customers'

'domain shit' --> 'domain shift'

---

> ### Author Response · Authors · 2020-11-17
> **Answer to Reviewer 2**
>
> Thank you for your feedback and for recognizing our theoretical contributions. We think that your concerns about experimental weaknesses can be addressed. We have responded to your comments below. We hope you reconsider your judgment.
>
> 1. Please note that when we discard a group of samples, the source samples are not discarded. The threshold only is used to pick the GMM samples. All the source and the training data points would still be used. As demonstrated in Eq. 6, none of the source samples are discarded. However, you have a fair point that we need further experiments to study the effect of the value of Tau. We tried to study this effect more accurately following your comment. Please refer to our answer to your next question. In short, it seems that we really don’t need to tune tau very carefully. As the theorem suggests, verified by the new experiment, a value that is reasonably close to 1 would work.
>
> 2.
> a) Please note that we did not mean to compare methods that use 5000 dimension vectors with the methods 30000 at all. As you pointed out, doing so is unreasonable. We regret not to mention this more explicitly in the paper. We clarified this point in the paper and changed the table to avoid this impression. What we meant was to compare the methods that use the same dimension against each other. We explored the literature for comparison and since TAT had reported results with 30000 features, we wanted to be complete in our Table. Unfortunately, the rest of the methods do not report their performance for the 30000 dimension so we couldn't include them.
>
> b) As pointed out by our theorem, the threshold should be close to 1 to make the upperbound in Eq 7 to be tight. So, we previously set it to be 0.95 following this intuition but we agree it was an arbitrary choice which could be done systematically according to your suggestion.
> c) Thank you for your suggestion. In response to your comment, we performed a new experiment and updated the paper using a curve. Please check the paper for the updated experiment. We hope adding this graph addresses your concern and convinces you that turning tau is not a bottleneck for our algorithm.
>
> 3. Thank you for pointing out the typos. We corrected them in the paper. We will make sure to have the paper proofread by a professional editor in our organization for the camera-ready version.

---

> > ### Comment · AnonReviewer2 · 2020-11-24
> > **Response**
> >
> > Thanks for addressing my concerns.  I think the current submission still needs further work. From Figure 3, the accuracy seems very sensitive to the value of \tau after 0.8, which is considered to be a bottleneck.

---

> > > ### Author Response · Authors · 2020-11-24
> > > **Performance dependence on the confidence parameter**
> > >
> > > Thank you for reading our rebuttal. It is correct that the performance depends on the value for parameter \tau a, but based on both our theoretical analysis and experimental exploration following your comment, we demonstrated that we need to set $\tau \approx 1$ and then the performance of our algorithm will in its best zone. Most algorithms have parameters that need to be tuned. In our case, we have demonstrated how to set the value. Could you specify why this is a bottleneck for our algorithm? In other words, what is your suggestion to improve the draft further?
> > >
> > > Thank you,
> > > our team

---

> > > > ### Comment · AnonReviewer2 · 2020-11-24
> > > > **Response**
> > > >
> > > > Thanks for your response. I see that the method need to set $\tau \approx 1$ and then the performance will in its best zone. However, for the experiment, you should follow some basic criterions to set the important parameters, e.g., tuning $\tau$ according to the validation data set.  In the method, as an important parameter, $\tau$ can be 0.99, 0.999 or even 0.9999 to achieve better performance. Moreover, from Figure 3, the ratio of slope of the line close to 1.0 seems very large and prove the sensitivity of $\tau$ in the method.

---

> > > > > ### Author Response · Authors · 2020-11-24
> > > > > **Performance dependence on the confidence parameter**
> > > > >
> > > > > We would like to thank you to engage in conversation and provide the chance to us to answer. We wanted to add that, tuning  $\tau$  according to the validation split is definitely possible and we can easily tune $\tau$ further according to an independent validation split for a camera-ready version. But we don't think it will change the outcome much. We agree that finding whether $\tau = 0.99$ or $\tau = 0.999$ is the optimal value can change the outcome but it will likely be a minor improvement. More importantly, our results are already competitive as can be seen in our table.  Please note that for any parameter in an algorithm, you can ask the same question, and tuning is done to some degree of accuracy in most cases.
> > > > >
> > > > > Finally, we do not want to insist on changing your opinion because, in the end, you have the absolute right to have an independent opinion. However, we think tuning $\tau$ is a minor issue for our algorithm.
> > > > >
> > > > > Thank you again,
> > > > > Our team

---

### Official Review · AnonReviewer1 · 2020-10-28
**AnonReviewer1**

**Rating:** 5
**Confidence:** 4

**Review:**

This paper proposes a new domain adaptation SAUM method that learns a large margin classifier between different classes for cross-domain sentiment analysis. The key idea is to train a domain-agnostic embedding space based on learning a prototypical source distribution with GMM, which is then used to align domain distributions via SWD minimization. Experimental results on Amazon multi-domain review dataset empirically demonstrate that the proposed SAUM method can outperform the state-of-the-art domain adaptation baselines. Moreover, detailed theoretical and empirical analysis are provided to demonstrate the effectiveness of the proposed method.

Strengths

1.The paper is reasonably well-written and structured.

2.The proposed SAUM method is technically sound.

3.The proposed SAUM method can outperform state-of-the-art domain adaptation methods.

4.Theoretical analysis and empirical analysis and sufficient and convinced.

Weaknesses

1.For domain adaptation in the NLP field, powerful pre-trained language models, e.g., BERT, XLNet, can overcome the domain-shift problem to some extent. Thus, the authors should be used as the base encoder for all methods and then compare the efficacy of the transfer parts instead of the simplest n-gram features.

2.The whole procedure is slightly complex. The author formulates the prototypical distribution as a GMM, which has high algorithm complexity. However, formal complexity analysis is absent. The author should provide an analysis of the time complexity and training time of the proposed SAUM method compared with other baselines. Besides, a statistically significant test is absent for performance improvements.

3.The motivation of learning a large margin between different classes is exactly discriminative learning, which is not novel when combined with domain adaptation methods and already proposed in the existing literature, e.g.,
Unified Deep Supervised Domain Adaptation and Generalization, Saeid et al., ICCV 2017.
Contrastive Adaptation Network for Unsupervised Domain Adaptation, Kang et al.,  CVPR 2019
Joint Domain Alignment and Discriminative Feature Learning for Unsupervised Deep Domain Adaptation, Chen et al., AAAI 2019.

However, this paper lacks detailed discussions and comparisons with existing discriminative feature learning methods for domain adaptation.

4.The unlabeled data (2000) from the preprocessed Amazon review dataset (Blitzer version) is perfectly balanced, which is impractical in real-world applications. Since we cannot control the label distribution of unlabeled data during training, the author should also use a more convinced setting as did in
Adaptive Semi-supervised Learning for Cross-domain Sentiment Classification, He et al., EMNLP 2018, which directly samples unlabeled data from millions of reviews.

5.The paper lacks some related work about cross-domain sentiment analysis, e.g.,
End-to-end adversarial memory network for cross-domain sentiment classification, Li et al., IJCAI 2017
Adaptive Semi-supervised Learning for Cross-domain Sentiment Classification, He et al., EMNLP 2018
Hierarchical attention transfer network for cross-domain sentiment classification, Li et al., AAAI 18


Questions:

1.Have the authors conducted the significance tests for the improvements?

2.How fast does this algorithm run or train compared with other baselines?

---

> ### Author Response · Authors · 2020-11-17
> **Answer to Reviewer 1**
>
> Thank you for your feedback. We are pleased that the reviewer recognizes the theoretical contribution of our work and has found the experiments to be convincing for the common setting in the literature. It seems to the authors that you have a positive assessment of the core idea of the paper and your concerns are more about some lacking aspects that we have tried address. We have responded to the points you raised about the weaknesses below. We hope you reconsider your initial review.
>
> 1. We agree with your statement that using pre-trained models such as BERT can mitigate domain-shift considerably. But please note that in the common domain adaptation scenario that we explore in this work, we are interested in transferring knowledge from a single “source domain” to a “target domain”. Models similar to BERT or XLNet need a very large corpus for pretraining. This means that we are relying on an additional huge “source domain” that is used to train these models. Hence, performing experiments with these models goes against the goal of domain adaptation. We agree however that these models address the challenge of domain-shift in many applications but please note that even for a method as good as BERT, domain-shift can be problematic if we test the model on domains that are very different from the training corpora. To sum up, we think using BERT or XLNet features are not fair for domain adaptation unless we test on a domain where domain-shift is a real challenge.
>
> 2. Learning a GMM typically has a high algorithmic complexity because iterative algorithms like EM must be used. But this is not the case for our problem. The major reason is that we learn GMM for a labeled dataset. For this reason, not only it is a much simpler task but providing exact complexity analysis is possible. We have updated the paper and added complexity analysis. Please refer to the paper Appendix to see our analysis. In short, we think that the complexity of learning GMM actually is not high compared to other computations, e.g., backpropagation.
>
> 3. Unfortunately, it is not easy to address a comment about novelty. We acknowledge the right of the reviewer to measure the subjective notion of “novelty” but we respectfully ask the reviewer to compare all of these four papers once more side-by-side and check whether our method is similar to these methods? We agree that all four methods try to make the embedding space discriminative. However, these works develop dramatically different algorithms to make the embedding space discriminative. Please note that two of these papers are published in 2019 after the 2017 paper. The 2019 papers have been assessed to be novel both by the reviewers and also the community, evident from their citation level.  These works were not discarded because the idea of “making the embedding space discriminative” had been used in 2017. Similarly, we think that our novelty is to propose a completely new approach to learn a discriminative embedding space, despite the fact that prior works have benefited from this strategy. Finally, none of the above methods address an NLP problem and our work might claim novelty in that it focuses on the less explored area of domain adaptation for NLP.
>
> 4. This is a fair concern and we agree with your statement. It is also a great suggestion for exploration in the area of domain adaptation. To address this comment, we performed an additional experiment by generating controlled imbalanced datasets from the  Amazon dataset. Please check the updated Appendix. In short, it seems that at least for the controlled experiments that we performed, label imbalance has not lead to significant performance degradation.
>
> 5. We referenced these works in our introduction.
> Questions:
>
> 1. We did not fully understand “the significance test” that you meant. Ablative studies are used to verify a similar notion in ML community in that they are used to verify whether the algorithm steps are indeed statistically effective. If you specify more what test you have in your mind, we hopefully can perform and update the draft further in the remaining time.
>
> 2. We measured the running time for our algorithm on our GPU cluster which is equipped with four GeForce RTX 2080 GPUs. Running the full pipeline takes 18 minutes on average. Existing works in the literature have not reported their running time so comparison against them is not straightforward, particularly due to the effect of hardware. Also, please note that since training is performed once in domain adaptation, as opposed to areas like online learning, training time is not a metric of interest in domain adaptation literature and to the best of our knowledge, no prior UDA algorithm has reported this metric.

---

### Author Response · Authors · 2020-11-17
**Note to the reviewers**

Dear reviewers,

Thank you for your collective feedback. Our hope is that based on your comments, we can improve our work during the rebuttal period to reach an acceptable level. Given the nature of reviews, we think this goal is reasonably possible. We have revised the paper text and our code according to the points you raised. In the remaining time, we hope to engage in more discussion to address the concerns.

---

### Decision · Program_Chairs · 2021-01-07
**Final Decision**

**Decision:**

Reject

**Comment:**

In this paper, the authors proposed a large-margin-based domain adaptation method for cross-domain sentiment analysis.

The idea of developing a large-margin-based method for domain adaptation is not new. Though the proposed method contains some new ideas,  the difference between the proposed method and the existing large-margin based methods needs to be discussed and studied empirically.  In addition, the experimental results are not convincing: some related baselines are missing and experiments need to be conducted on more datasets.

Though the authors did provide long responses to each reviewer, after a lot of discussions, the reviewers still find that their concerns are not well addressed.

Therefore, this paper is not ready to be published in ICLR based on its current shape.